# Eosinophils and the Efficacy of Immune Checkpoint Inhibitors Across Multiple Cancers: A Retrospective Study

**DOI:** 10.3390/biomedicines13123029

**Published:** 2025-12-10

**Authors:** Risako Suzuki, Ryotaro Ohkuma, Makoto Watanabe, Emiko Mura, Toshiaki Tsurui, Nana Iriguchi, Tomoyuki Ishiguro, Yuya Hirasawa, Go Ikeda, Masahiro Shimokawa, Hirotsugu Ariizumi, Yutaro Kubota, Kiyoshi Yoshimura, Shinichi Kobayashi, Takuya Tsunoda, Atsushi Horiike, Mayumi Tsuji, Yuji Kiuchi, Tatsunori Oguchi, Satoshi Wada

**Affiliations:** 1Department of Medical Oncology, Department of Medicine, School of Medicine, Showa Medical University, Tokyo 142-8666, Japan; 2Department of Clinical Diagnostic Oncology, Clinical Research Institute for Clinical Pharmacology and Therapeutics, Showa Medical University, Tokyo 157-8577, Japan; 3Department of Pharmacology, Showa Medical University Graduate School of Medicine, Tokyo 142-8555, Japan; 4Comprehensive Cancer Information Center, Showa Medical University, Tokyo 142-8555, Japan; 5Pharmacological Research Center, Showa Medical University, Tokyo 142-8555, Japan; 6Clinical Research Institute for Clinical Pharmacology and Therapeutics, Showa Medical University, Tokyo 157-8577, Japan; 7Department of Clinical Immuno Oncology, Clinical Research Institute for Clinical Pharmacology and Therapeutics, Showa Medical University, Tokyo 157-8577, Japan

**Keywords:** eosinophil, immune checkpoint inhibitor, biomarker, NLR, overall survival, progression-free survival, retrospective study

## Abstract

**Background/Objectives**: Immune checkpoint inhibitors (ICIs) have transformed cancer therapy; however, their efficacy varies among patients. Eosinophils have been reported as prognostic markers in chemotherapy-treated pancreatic cancer; however, these studies are limited mainly to single cancer types and relatively small cohorts. Therefore, we aimed to examine the relationship between eosinophil count and the effectiveness of ICIs across various cancer types. **Methods**: We retrospectively analyzed 138 patients treated with ICI monotherapy, ICI plus chemotherapy, or chemotherapy alone at our institution between December 2015 and September 2024. Peripheral blood parameters, including eosinophil counts and albumin levels, were collected at baseline and after two cycles of treatment. Associations between overall survival (OS) and progression-free survival (PFS) were assessed using Kaplan–Meier analysis and Cox proportional hazards regression. **Results**: In the ICI monotherapy group, patients with higher baseline eosinophil counts had significantly longer OS (HR 0.26, *p* = 0.007) and PFS (HR 0.30, *p* = 0.005). No significant associations were observed between the ICI plus chemotherapy and chemotherapy-alone groups. Changes in eosinophil counts between baseline and after two cycles were not associated with outcomes in any group. **Conclusions**: Baseline eosinophil counts were significantly associated with survival outcomes in patients receiving ICI monotherapy and may serve as a promising predictive biomarker.

## 1. Introduction

Immune checkpoint inhibitors (ICIs) targeting programmed cell death-1 (PD-1), programmed death ligand-1 (PD-L1), and cytotoxic T-lymphocyte-associated protein 4 (CTLA-4) have revolutionized cancer therapy by restoring antitumor immunity. Immune checkpoint inhibitors (ICIs) provide durable clinical responses in a subset of patients with cancer. Although programmed death ligand 1 (PD-L1) expression is used as a biomarker, its predictive power is limited, necessitating the identification of more accurate and broadly applicable indicators. Biomarkers derived from tumor tissue are often limited by sampling heterogeneity, cost, and time-consuming analysis. Consequently, attention has shifted toward peripheral blood indicators that can capture real-time systemic immunity. Other markers, such as the neutrophil-to-lymphocyte ratio (NLR) and serum albumin levels, have been investigated as predictors, but none have been universally validated [1,2,3,4,5,6]. Moreover, the mechanisms underlying ICI efficacy remain unclear. The antitumor activity of ICIs largely depends on the pre-existing immune landscape of the host. Tumors with abundant immune infiltration, particularly by activated CD8^+^ T cells, tend to respond more favorably to checkpoint blockade. However, quantifying tumor immune contexture requires invasive biopsies and is often hindered by spatial heterogeneity. Peripheral blood, in contrast, provides a dynamic and easily obtainable window into systemic immunity. Parameters such as lymphocyte subsets, monocytes, and eosinophils can mirror the tumor microenvironment and offer real-time insights into immune activation. Recent studies have emphasized the prognostic importance of peripheral immune signatures, including ratios such as lymphocyte-to-monocyte or platelet-to-lymphocyte, reflecting the host’s inflammatory and nutritional status. These findings suggest that comprehensive immune profiling of peripheral blood may serve as a surrogate for tissue-based immunological assessment, particularly in patients where tumor sampling is limited or unfeasible.

Eosinophils, traditionally recognized for their roles in allergic responses and parasitic infections, have recently attracted increasing attention in tumor immunology. Several studies on non-small-cell lung cancer, urothelial carcinoma, and melanoma have linked higher baseline eosinophil counts to improved survival in patients treated with ICIs [7,8,9,10]. Additionally, associations between eosinophils and immune-related adverse events (irAEs) have been described [9,10,11]. Eosinophils have also been reported as prognostic markers in chemotherapy-treated pancreatic cancer, underscoring their broad clinical relevance [12]. As demonstrated in the studies mentioned above, several studies have highlighted associations between peripheral immune cell composition and therapeutic efficacy. Incorporating such easily measurable hematological indicators into clinical evaluation may broaden the ability to predict responses to immune checkpoint inhibitors (ICIs) beyond conventional tissue-based biomarkers. However, these studies are mainly limited to single cancer types and relatively small cohorts.

We hypothesized that baseline eosinophil levels, reflecting innate immune readiness, could serve as an integrative biomarker predicting ICI responsiveness irrespective of tumor origin. To test this hypothesis, we aimed to clarify the association between baseline eosinophil counts and ICI efficacy across multiple cancer types.

## 2. Materials and Methods

### 2.1. Study Design and Patients

We retrospectively reviewed the data of 138 patients treated at our hospital between December 2015 and September 2024. Eligible cancer types included lung, gastric, esophageal, colorectal, melanoma, and pancreatic cancers. Patients were categorized into three groups: ICI monotherapy (*n* = 37), ICI plus chemotherapy (*n* = 48), and chemotherapy alone (*n* = 53) (Table 1). Informed consent for participation in the study was obtained from all patients.

### 2.2. Data Collection

Peripheral blood data obtained at baseline and after two cycles included white blood cell counts, eosinophils (both absolute and percentage), lymphocytes, monocytes, and albumin levels. Performance status (PS), sex, and occurrence of immune-related adverse events (irAEs; classified according to CTCAE) were also recorded. To evaluate whether post-treatment changes in eosinophil counts were associated with clinical outcomes, we assessed eosinophil dynamics between before treatment initiation and after two cycles. Eosinophil “increase” was defined as a higher eosinophil count at the beginning of cycle 3 compared with the value measured immediately before cycle 1, whereas “decrease” was defined as a lower value at cycle 3 than at baseline.

### 2.3. Outcomes and Statistical Analysis

The primary endpoints were overall survival (OS) and progression-free survival (PFS). Survival distributions were estimated using the Kaplan–Meier method and compared using the log-rank test. Hazard ratios (HRs) and 95% confidence intervals (CIs) were calculated using Cox proportional hazards regression models.

Variables were dichotomized based on their median values, except for PS, irAEs, and sex. PS was categorized into favorable (0–1) and unfavorable (2–3) groups. IrAEs were classified as present or absent, and sex was analyzed as male or female.

Variables with a *p* value < 0.10 in the univariable analysis were subsequently included in the multivariable Cox regression. In addition, clinically relevant factors, including age, sex, and PS, were incorporated into the multivariate model, regardless of their statistical significance in the univariate analysis. Statistical significance was set at *p* < 0.05.

## 3. Results

### 3.1. Baseline Eosinophil Counts and Survival

In the ICI monotherapy group, the median baseline eosinophil count was 92.3/µL. Patients with counts above this cutoff had significantly longer OS (HR 0.26, 95% CI 0.09–0.79, *p* = 0.007) (Figure 1) and PFS (HR 0.30, 95% CI 0.12–0.73, *p* = 0.005) (Figure 2). No significant associations were observed between the ICI plus chemotherapy and chemotherapy-alone groups.

In the ICI monotherapy group, 23 patients were treated with anti-PD-1 therapy and 14 patients were treated with anti-PD-1 and anti-CTLA-4 combination therapy. Among patients treated with anti-PD-1 monotherapy, both OS and PFS were significantly longer in the high-baseline-eosinophil group (OS: *p* = 0.01; PFS: *p* = 0.01). In the anti-PD-1 + anti-CTLA-4 subgroup, statistical significance was not reached; however, a similar trend toward better outcomes in patients with higher baseline eosinophil counts was observed. No patients in our cohort received anti-PD-L1 monotherapy. In the ICI + chemotherapy group, 38 patients were treated with Anti-PD-1 combined with chemotherapy, 10 patients were treated with Anti-PD-L1 combined with chemotherapy, In the ICI plus chemotherapy group, 38 patients received anti-PD-1 and 10 patients received anti-PD-L1. For both anti-PD-1 and anti-PD-L1 combined with chemotherapy, neither OS nor PFS showed significant associations with baseline eosinophil counts (anti-PD-1: OS *p* = 0.15, PFS *p* = 0.19; anti-PD-L1: OS *p* = 0.66, PFS *p* = 0.26), which is consistent with the findings observed for the overall combination-therapy cohort.

In the ICI monotherapy group, the cancer-type distribution was as follows: lung cancer (*n* = 16), esophageal cancer (*n* = 14), melanoma (*n* = 6), and gastric cancer (*n* = 1). Because only a single gastric cancer case was included, subtype-specific analyses were not feasible and therefore were not reported. We further examined the association between baseline absolute eosinophil count and survival within each cancer subtype. The *p*-values for OS and PFS were as follows: esophageal cancer (OS *p* = 0.28, PFS *p* = 0.30), lung cancer (OS *p* = 0.05, PFS *p* = 0.02), and melanoma (OS *p* = 0.19, PFS *p* = 0.19). Although statistical significance was observed only in lung cancer subgroup, all subtypes demonstrated a consistent trend in which higher baseline eosinophil counts were associated with more favorable clinical outcomes.

### 3.2. Post-Treatment Eosinophil Changes

To evaluate whether changes in eosinophil counts after treatment were associated with clinical outcomes, we compared eosinophil levels before treatment initiation and after two cycles. In the ICI monotherapy group, 22 patients (59.5%) showed an increase in eosinophil count, whereas 15 patients (40.5%) showed a decrease. In the ICI plus chemotherapy group, eosinophil counts increased in 14 patients (29.2%) and decreased in 34 patients (70.8%). In the chemotherapy-alone group, 18 patients (34.0%) exhibited an increase and 35 patients (66.0%) showed a decrease. Notably, these post-treatment eosinophil changes were not significantly associated with OS or PFS in any treatment cohort (Figure 3).

### 3.3. Univariable Analysis

Higher baseline eosinophil counts were significantly associated with improved OS (HR 0.26, 95% CI 0.087–0.788, *p* = 0.01) and PFS (HR 0.30, 95% CI 0.123–0.733, *p* = 0.01). A Low NLR was also associated with better OS (HR 3.31, 95% CI 1.399–7.832, *p* = 0.01) and PFS (HR 1.40, 95% CI 1.402–7.205, *p* = 0.01). Albumin levels showed a trend toward significance for OS (*p* = 0.06) and PFS (*p* = 0.10), although the differences were not significant. Other variables, including PS, sex, age, and irAEs, were not significantly associated with survival rates (Table 2).

### 3.4. Multivariate Analysis

Variables with *p* < 0.10 in the univariable analysis (baseline eosinophil count, albumin, and NLR) were entered into the multivariable Cox model, together with clinically relevant covariates (age, sex, and PS). Higher baseline eosinophil counts were independently associated with longer OS (HR 0.31, 95% CI 0.097–0.981, *p* = 0.03) and PFS (HR 0.32, 95% CI 0.119–0.838, *p* = 0.01). Although the NLR was not statistically significant (OS: HR 3.09, 95% CI 0.939–10.148, *p* = 0.06; PFS: HR 2.65, 95% CI 0.910–7.704, *p* = 0.07), the observed trends suggest that NLR is an important prognostic factor in patients treated with ICIs. Serum albumin levels were not significant in the multivariate analysis. Other factors, including age, sex, and PS, were not significantly associated with survival rates (Table 3).

## 4. Discussion

This study demonstrated that higher baseline eosinophil counts were associated with prolonged OS and PFS in patients treated with ICI monotherapy, consistent with previous reports [7,8,9,10]. The cutoff value identified in our cohort was comparable to that described in previous studies. In contrast, no such association was observed in patients receiving ICIs in combination with chemotherapy or chemotherapy alone, suggesting that eosinophils are directly associated with the immune-mediated effects of ICIs. This observation suggests that eosinophils may function as immune effector cells involved in antigen presentation, T-cell priming, and recruitment of other immune subsets. Their presence before therapy may indicate a pre-activated immune microenvironment capable of mounting effective responses upon checkpoint blockade.

Although chemotherapy-induced bone marrow suppression may reduce eosinophil counts after treatment, it should not affect baseline eosinophil levels. Most combination regimens in this study included platinum-based agents. Platinum compounds have been reported to enhance CD8^+^ T-cell infiltration and increase the ratio of PD-L1^+^ immune cells, while simultaneously inducing immunosuppressive effects via TGF-β signaling [13,14]. Moreover, the immunologic interplay between cytotoxic agents and immune cells is complex. Certain chemotherapies can induce immunogenic cell death and enhance antigen presentation, whereas others suppress bone marrow progenitors. Disentangling these bidirectional effects remains a key research priority. The early crossing of the Kaplan–Meier curves between the combination and monotherapy groups may reflect the reduced early progression in the combination group. Although chemotherapy may attenuate the intrinsic efficacy of ICIs, its ability to suppress early progression may counterbalance this effect and contribute to improved overall outcomes. The careful selection of cytotoxic agents with synergistic immunomodulatory activities is important for optimizing future treatment strategies.

Although NLR did not reach statistical significance in our multivariate analysis, the observed trends suggest that it continues to contribute to survival outcomes. These findings are consistent with previous reports demonstrating that a lower NLR is associated with improved prognosis in patients receiving ICIs across multiple cancer types. The NLR reflects systemic inflammation and immune balance, and an elevated NLR has been linked to an immunosuppressive tumor microenvironment (TME). Therefore, although NLR did not emerge as an independent prognostic factor in our cohort, our results support its potential role as a clinically relevant biomarker, consistent with prior evidence [1,2,3,4,5,6]. Combining eosinophil counts with NLR or albumin could provide a multifactorial immune index integrating inflammation, nutrition, and baseline immunity. Such indices may help stratify patients for ICI monotherapy versus combination regimens and optimize timing for treatment initiation.

In addition to their potential as biomarkers, eosinophils play diverse immunological roles in the TME. Recent studies have shown that eosinophils can infiltrate tumors and release granule proteins, such as eosinophil cationic protein and eosinophil peroxidase, as well as cytokines, including TNF-α, IL-4, and IL-13, which can induce direct tumor cell cytotoxicity and promote CD8^+^ T-cell and dendritic cell activation [15,16,17]. Moreover, eosinophils may remodel the extracellular matrix to facilitate lymphocyte infiltration and enhance the response to immune checkpoint blockade [18,19]. Conversely, eosinophils may exert pro-tumorigenic effects by secreting angiogenic and tissue-remodeling factors, depending on the cytokine milieu (e.g., IL-5, IL-33, and GM-CSF) within the TME [19,20].

In the context of ICI therapy, several studies have demonstrated that increased peripheral eosinophil counts following treatment correlate with activated antitumor immunity and, in some cases, with immune-related adverse events (irAEs) [21,22,23]. Thus, eosinophil dynamics may reflect the systemic immune activation triggered by ICIs. These findings support the hypothesis that eosinophils act as functional modulators of immune responses during checkpoint inhibition, in addition to serving as peripheral biomarkers.

Collectively, our findings and previous evidence suggest that the baseline eosinophil count represents an integrated indicator of host immune readiness. Beyond the prognostic perspective, our findings raise important implications for clinical practice and trial design. Integrating eosinophil assessment into pretreatment evaluation could help identify patients most likely to benefit from ICI monotherapy, thereby avoiding unnecessary exposure to cytotoxic chemotherapy. Furthermore, dynamic monitoring of eosinophil counts during treatment might serve as an early marker of immune activation, preceding radiographic changes. Evaluating eosinophil dynamics, along with other immune parameters such as NLR and albumin, may improve the prediction of ICI efficacy and provide insight into the complex immunological mechanisms governing antitumor responses.

### Limitations

This study has a few limitations. First, although multiple cancer types were included in the analysis, the number of patients in each subtype—particularly melanoma and gastric cancer—was small, which limited the statistical power of the subtype-specific survival analyses. In the ICI monotherapy cohort, gastric cancer consisted of only one case and therefore could not be evaluated separately. Second, PS was assessed by both physicians and nurses, potentially introducing interobserver variability; most patients had a PS of 1, limiting stratification by this variable. Finally, the analysis was restricted to peripheral blood data, and eosinophil infiltration was not examined in tumor tissues. Another limitation of this study is its retrospective design, which may have introduced selection bias and unmeasured confounding factors. In addition, we did not evaluate other immune cell subsets, such as regulatory T cells or myeloid-derived suppressor cells, which may interact with eosinophils to shape the tumor immune microenvironment. Future prospective multicenter studies are needed to validate and expand upon our findings. Future studies should integrate tumor-local eosinophil analyses to clarify their role in ICI responses.

## 5. Conclusions

The findings of this study showed a significant association between baseline eosinophil count and OS and PFS in patients who received ICI monotherapy. In clinical practice, baseline eosinophil count could be easily adopted as part of routine pretreatment evaluations. In contrast, no such association was observed in patients treated with ICIs combined with chemotherapy or chemotherapy alone, suggesting that eosinophils are explicitly associated with the immune-mediated effects of ICIs.

## Figures and Tables

**Figure 1 biomedicines-13-03029-f001:**
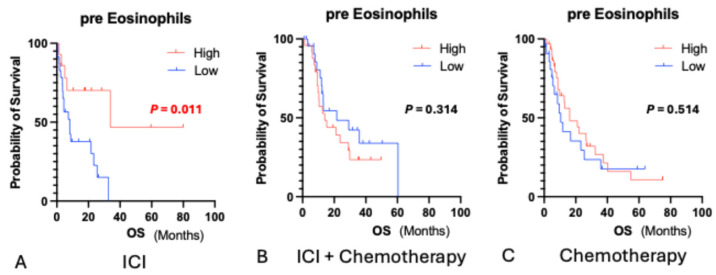
Overall survival according to baseline eosinophil count in patients. Kaplan–Meier survival curves for overall survival (OS) stratified by baseline eosinophil count (cutoff: 92.3/μL) in the ICI monotherapy, ICI plus chemotherapy, and chemotherapy alone groups. A significant survival benefit was observed only in the ICI monotherapy group for patients with higher eosinophil counts (HR: 0.33, 95%CI: 0.14–0.77, *p* = 0.01). No significant differences were detected between the combination and chemotherapy-only groups. Red font—statistically significant.

**Figure 2 biomedicines-13-03029-f002:**
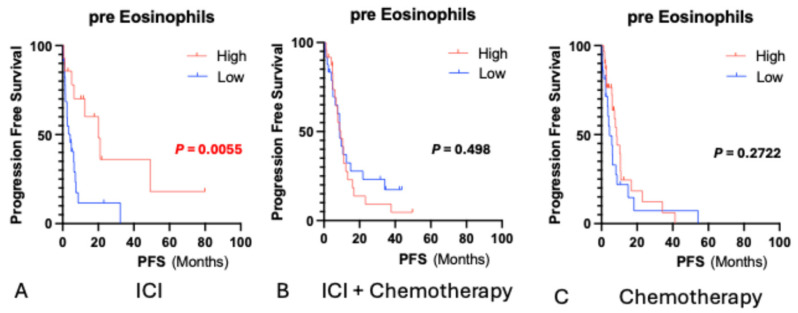
Progression-free survival according to baseline eosinophil count in patients. Kaplan–Meier survival curves for progression-free survival (PFS) stratified by baseline eosinophil count (cutoff: 92.3/μL) in the ICI monotherapy, ICI plus chemotherapy, and chemotherapy alone groups. Patients with higher eosinophil counts showed significantly longer PFS in the ICI monotherapy group (HR: 0.32, 95%CI: 0.15–0.72, *p* = 0.001), while no association was found in the other two groups. Red font—statistically significant.

**Figure 3 biomedicines-13-03029-f003:**
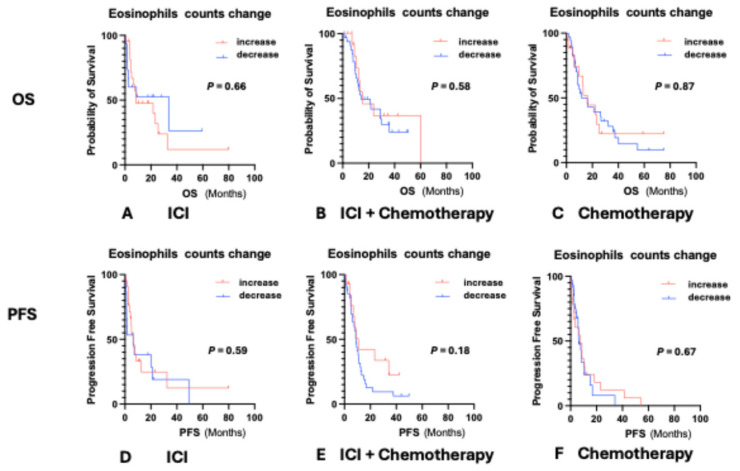
Changes in eosinophil counts between baseline and after two cycles in relation to OS and PFS across treatment groups. Changes in eosinophil counts between baseline and after two cycles were not significantly associated with OS (**A**–**C**) or PFS (**D**–**F**) in any treatment group. This finding suggests that dynamic eosinophil variations during early treatment do not reflect long-term outcomes, in contrast to the prognostic impact of baseline eosinophil levels.

**Table 1 biomedicines-13-03029-t001:** Baseline Characteristics of the Study Population.

Characteristic	ICI Monotherapy (*n* = 37)	ICI + Chemotherapy (*n* = 48)	Chemotherapy Alone (*n* = 53)	Total (*n* = 138)
Age, median (range)	70 (47–83)	71 (45–88)	72 (38–85)	71 (38–88)
Sex, *n* (%)				
Male	32 (86.5)	37 (77.0)	34 (64.2)	103 (74.6)
Female	5 (13.5)	11 (23.0)	19 (35.8)	35 (25.4)
Cancer type, *n* (%)				
Lung	16 (43.2)	14 (29.2)	1 (1.9)	31 (22.5)
Stomach	1 (2.7)	19 (39.6)	17 (32.1)	37 (26.8)
Esophageal	14 (37.8)	15 (31.3)	6 (11.3)	35 (25.4)
Colon	0	0	21 (39.6)	21 (15.2)
Melanoma	6 (16.2)	0	0	6 (4.3)
Pancreatic	0	0	5 (9.4)	5 (3.6)
Others	0	0	3 (5.7)	3 (2.2)
ECOG PS, *n* (%)				
0	7 (18.9)	10 (20.8)	3 (5.7)	20 (14.5)
1	24 (64.9)	35 (72.9)	44 (83.0)	103 (74.6)
2	3 (8.1)	2 (4.2)	4 (7.5)	9 (6.5)
3	3 (8.1)	1 (2.1)	2 (3.8)	6 (4.3)
Eosinophil count, median (/µL) (range)	92.3 (0–676)	104.4 (0–954)	132.6 (0–637)	103.2 (0–954)
Lymphocyte count, median (/µL) (range)	1270 (310–3260)	1205 (370–2860)	1170 (560–4600)	1205 (310–4600)
Albumin, median (g/dL) (range)	3.9 (1.5–4.5)	3.7 (1.7–4.5)	3.6 (2.1–4.5)	3.7 (1.5–4.5)
NLR, median	2.84 (0.63–11.8)	3.2 (1.2–15.7)	3.97 (0.78–16.0)	3.2 (0.63–16.0)
Change in eosinophil count, *n* (%) (before treatment → after 2 cycles)				
increase	22 (59.5)	14 (29.2)	18 (34)	54 (39)
decrease	15 (40.5)	34 (70.8)	35 (66)	84 (61)
irAE, *n* (% *)	21 (56.8)	25 (52.1)	–	46 (54.1 *)
Grade 1	4 (10.8)	8 (16.7)	–	12 (14.1 *)
Grade 2	7 (18.9)	2 (4.2)	–	15 (17.6 *)
Grade 3	8 (21.6)	9 (18.8)	–	17 (20.0 *)
Grade 4	1 (2.7)	0	–	1 (1.2 *)
Grade 5	1 (2.7)	0	–	1 (1.2 *)
None	16 (43.2)	23 (47.9)	–	39 (45.9 *)

* irAE incidence was evaluated among the ICI monotherapy and ICI plus chemotherapy groups only.

**Table 2 biomedicines-13-03029-t002:** Univariable analysis of factors associated with overall survival (OS) and progression-free survival (PFS).

Univariable Analysis				
Variable		Hazard Ratio	95% CI (Lower–Upper)	*p*-Value
Overall Survival (OS)				
Eosinophil count (Pre)	High vs. Low	0.26	0.087–0.788	0.01
NLR	High vs. Low	3.31	1.399–7.832	0.01
Albumin (Pre)	High vs. Low	0.44	0.193–1.020	0.06
ECOG PS	0–1 vs. 2–3	0.47	0.158–1.425	0.21
irAE	with irAE vs. without irAE	0.62	0.264–1.434	0.27
Sex	Male vs. Female	0.65	0.188–2.251	0.52
Age	High vs. Low	1.60	0.655–3.913	0.31
Progression-Free Survival (PFS)				
Eosinophil count (Pre)	High vs. Low	0.30	0.123–0.733	0.01
NLR	High vs. Low	1.40	1.402–7.205	0.01
Albumin (Pre)	High vs. Low	0.51	0.232–1.112	0.09
irAE	with irAE vs. without irAE	0.59	0.272–1.274	0.19
ECOG PS	0–1 vs. 2–3	1.63	0.551–4.840	0.40
Sex	Male vs. Female	1.09	0.326–3.659	0.89
Age	High vs. Low	1.20	0.536–2.703	0.66

**Table 3 biomedicines-13-03029-t003:** Multivariate Analysis of Prognostic Factors for OS and PFS in ICI Monotherapy.

Multivariable Analysis				
Variable		Hazard Ratio	95% CI (Lower–Upper)	*p*-Value
Overall Survival (OS)				
Eosinophil count (Pre)	High vs. Low	0.31	0.097–0.981	0.03
NLR	High vs. Low	3.09	0.939–10.148	0.06
Sex	Male vs. Female	0.26	0.053–1.259	0.11
ECOG PS	0–1 vs. 2–3	0.34	0.096–1.235	0.13
Age	High vs. Low	1.94	0.701–5.341	0.21
Albumin (Pre)	High vs. Low	0.51	0.163–1.621	0.25
irAE	with irAE vs. without irAE	0.68	0.251–1.820	0.44
Progression-Free Survival (PFS)				
Eosinophil count (Pre)	High vs. Low	0.32	0.119–0.838	0.01
NLR	High vs. Low	2.65	0.910–7.704	0.07
irAE	with irAE vs. without irAE	0.46	0.196–1.650	0.07
ECOG PS	0–1 vs. 2–3	0.48	0.142–1.629	0.26
Albumin (Pre)	High vs. Low	0.64	0.245–1.671	0.36
Age	High vs. Low	0.75	0.302–1.884	0.54
Sex	Male vs. Female	0.71	0.349–5.700	0.64

## Data Availability

The datasets generated and/or analyzed during the current study are not publicly available due to patient privacy and institutional regulations but are available from the corresponding author on reasonable request.

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
