# Peer review of "Eosinophils and the Efficacy of Immune Checkpoint Inhibitors Across Multiple Cancers: A Retrospective Study"

_biomedicines, 2025, doi:10.3390/biomedicines13123029_

Round 1
Reviewer 1 Report
Comments and Suggestions for Authors
The manuscript by Risak Suzuke et al. demonstrated an interesting association between higher eosinophil counts in the blood and better response to ICI monotherapy, suggesting eosinophil count as a potential biomarker for ICI therapy. As the authors acknowledge, the study has limitations, including a small patient cohort and the lack of additional tumor histopathology information. A more careful characterization of tumor subtypes in relation to ICI response, or an increase in sample size, is needed to strengthen the conclusions. Below is a list of issues that need to be addressed or clarified:
- Clarification of ICI therapy used: Which specific ICI therapy was used in the study—anti-PD1, anti-PD-L1, or anti-CTLA4? If multiple therapies were used, the data should be stratified accordingly to show the correlation between eosinophil count and different treatments.
- Cancer subtype information: Lung cancer, esophageal cancer, and melanoma were the primary cancer subtypes included in the study for ICI monotherapy. To assess whether there is a correlation between AEC (absolute eosinophil count) and ICI response across all cancer types, Figures 1A and 1B need to provide additional information, including median survival time and p-values for each subtype.
- Threshold for eosinophil changes: In Figure 3, the thresholds for "UP" and "DOWN" changes in eosinophil counts need to be clearly defined. Since eosinophil assays have technical variability and normal fluctuations, the thresholds should account for these factors and reflect only significant changes. Additionally, what percentage of patients showed an increase or decrease in AEC post-treatment?
- Chemotherapy-induced immunogenic cell death: The authors mention that certain chemotherapies can induce immunogenic cell death. Was this observed in the study? What percentage of patients showed downregulation of eosinophil counts upon receiving ICI + chemotherapy or chemotherapy alone?
- Typographical errors: There is a typo in Figures 1, 2, and 3 where "chemotherapy" is misspelled as "chemothrapy."
Author Response
1Clarification of ICI therapy used: Which specific ICI therapy was used in the study—anti-PD1, anti-PD-L1, or anti-CTLA4? If multiple therapies were used, the data should be stratified accordingly to show the correlation between eosinophil count and different treatments.
Thank you very much for this important comment. We have now clarified the specific types of immune checkpoint inhibitors administered in each treatment group, and corresponding stratified analyses have been added as Supplementary Figures in the revised manuscript.
In the ICI monotherapy group, the following agents were used:
・Anti–PD-1 therapy: 23 patients
・Anti–PD-1 + anti–CTLA-4 combination therapy: 14 patients
Among patients treated with anti–PD-1 monotherapy, both OS and PFS were significantly longer in the high–baseline eosinophil group (OS: P = 0.01; PFS: P = 0.01).
In the anti–PD-1 + anti–CTLA-4 subgroup, statistical significance was not reached; however, a similar trend toward better outcomes in patients with higher baseline eosinophil counts was observed.
No patients in our cohort received anti–PD-L1 monotherapy.
In the ICI + chemotherapy group, the following agents were administered:
・Anti–PD-1: 38 patients
・Anti–PD-L1: 10 patients
For both anti–PD-1 and anti–PD-L1 combined with chemotherapy, neither OS nor PFS showed significant associations with baseline eosinophil counts (anti–PD-1: OS P = 0.15, PFS P = 0.19; anti–PD-L1: OS P = 0.66, PFS P = 0.26), which is consistent with the findings observed for the overall combination-therapy cohort.
These details have now been added to the Results section, and stratified analyses for each ICI category are now presented in the updated Supplementary Figures.
2 Cancer subtype information: Lung cancer, esophageal cancer, and melanoma were the primary cancer subtypes included in the study for ICI monotherapy. To assess whether there is a correlation between AEC (absolute eosinophil count) and ICI response across all cancer types, Figures 1A and 1B need to provide additional information, including median survival time and p-values for each subtype.
We appreciate the reviewer’s insightful comments.
In the ICI monotherapy cohort, the distribution of cancer subtypes was as follows:
・Lung cancer: 16 patients
・Esophageal cancer: 14 patients
・Melanoma: 6 patients
・Gastric cancer: 1 patient
Because only a single gastric cancer case was included, subtype-specific analyses were not feasible and therefore were not reported.
The median OS and PFS for each cancer subtype were as follows:
・Lung cancer: OS 9.2 months, PFS 6.5 months
・Esophageal cancer: OS 9.3 months, PFS 6.4 months
・Melanoma: OS 11.0 months, PFS 4.0 months
We further examined the association between baseline absolute eosinophil count and survival within each cancer subtype:
・Esophageal cancer: OS p = 0.28, PFS p = 0.30
・Lung cancer: OS p = 0.05, PFS p = 0.02
・Melanoma: OS p = 0.19, PFS p = 0.19
Although statistical significance was observed only in lung cancer subgroup, all subtypes demonstrated a consistent trend in which higher baseline eosinophil counts were associated with more favorable clinical outcomes.
These results have now been incorporated into the revised manuscript and added as a supplementary figure. In addition, we have expanded the Limitations section to acknowledge that the relatively small sample size, especially for melanoma and gastric cancer, may have limited the statistical power of subtype-specific analyses.
3. Threshold for eosinophil changes: In Figure 3, the thresholds for "UP" and "DOWN" changes in eosinophil counts need to be clearly defined. Since eosinophil assays have technical variability and normal fluctuations, the thresholds should account for these factors and reflect only significant changes. Additionally, what percentage of patients showed an increase or decrease in AEC post-treatment?
Thank you for this important comment. We have now clarified the definitions of eosinophil “increase” and “decrease,” and we have replaced the labels “UP/DOWN” with “increase/decrease” in the revised figure for improved clarity.
Definition of eosinophil change:
・Increase: Eosinophil count at the beginning of cycle 3 was higher than the value measured immediately before cycle 1.
・Decrease: Eosinophil count at the beginning of cycle 3 was lower than the value measured immediately before cycle 1.
Number (percentage) of patients in each category:
ICI monotherapy: Increase 22 (59.5%), Decrease 15 (40.5%)
ICI + chemotherapy: Increase 14 (29.2%), Decrease 34 (70.8%)
Chemotherapy alone: Increase 18 (34.0%), Decrease 35 (66.0%)
These data have been added to the revised manuscript. The definitions have also been incorporated into the Methods section, and Table1 has been updated accordingly.
4. Chemotherapy-induced immunogenic cell death: The authors mention that certain chemotherapies can induce immunogenic cell death. Was this observed in the study? What percentage of patients showed downregulation of eosinophil counts upon receiving ICI + chemotherapy or chemotherapy alone?
Thank you for raising this important question.
As our study focused on peripheral eosinophil dynamics, we did not directly assess tumor-level biomarkers of immunogenic cell death (ICD). Therefore, our dataset does not allow us to determine whether ICD occurred in individual patients.
However, to indirectly evaluate treatment-related immune effects, we analyzed changes in eosinophil counts before and after therapy. The proportions of patients showing a decrease in eosinophil counts were as follows:
ICI monotherapy: 15 patients (40.5%)
ICI + chemotherapy: 34 patients (70.8%)
Chemotherapy alone: 35 patients (66.0%)
These findings are consistent with the known bone-marrow suppressive effects of cytotoxic chemotherapy and support the interpretation that eosinophil dynamics may more accurately reflect the immune-mediated effects in the context of ICI monotherapy.
- Typographical errors: There is a typo in Figures 1, 2, and 3 where "chemotherapy" is misspelled as "chemothrapy."
Thank you for pointing this out.
The typographical error (“chemothrapy”) appearing in Figures 1–3 has been corrected in the revised version of the manuscript.
Reviewer 2 Report
Comments and Suggestions for Authors
Suzuki et al aim to offer intriguing insights into the emerging links between eosinophilia, checkpoint inhibition and patient outcomes. It is clear, however, that eosinophil counts (+/- immune checkpoint inhibitors) are reflective of both effective tumour rejection and high overall survival as well as adverse events and mortality, and these outcomes depend on cancer type. It's not clear to me if the authors have grouped all the different tumour types they sampled together or are the figures showing just one tumour type? I would prefer it, if the authors were to either stratify their data by tumour anatomical site or beef up their methods section explaining in greater detail how the patients/samples were grouped prior to analysis.
Author Response
It is clear, however, that eosinophil counts (+/- immune checkpoint inhibitors) are reflective of both effective tumour rejection and high overall survival as well as adverse events and mortality, and these outcomes depend on cancer type. It's not clear to me if the authors have grouped all the different tumour types they sampled together or are the figures showing just one tumour type? I would prefer it, if the authors were to either stratify their data by tumour anatomical site or beef up their methods section explaining in greater detail how the patients/samples were grouped prior to analysis.
We thank the reviewer for this insightful comment. As requested, we performed additional stratified analyses by cancer type within the ICI monotherapy cohort. The distribution of tumor types was as follows: lung cancer (n = 16), esophageal cancer (n = 14), melanoma (n = 6), and gastric cancer (n = 1). Because gastric cancer included only a single patient, it was excluded from subtype-specific survival analyses.
The median overall survival (OS) and progression-free survival (PFS) for each cancer type were as follows:
・Lung cancer: OS 9.3 months, PFS 6.5 months
・Esophageal cancer: OS 9.2 months, PFS 6.4 months
・Melanoma: OS 11.1 months, PFS 4.0 months
We then evaluated whether baseline eosinophil count was associated with survival within each tumor type. The results were:
・Esophageal cancer: OS P = 0.28, PFS P = 0.30
・Lung cancer: OS P = 0.05, PFS P = 0.02
・Melanoma: OS P = 0.19, PFS P = 0.19
Although statistical significance was observed only in lung cancer subgroup, similar favorable trends were observed across tumor types. These analyses and findings have been incorporated into the revised manuscript, and a new supplementary figure (Supplementary Figure 3) has been created to illustrate the subtype-specific results.
We appreciate the reviewer’s suggestion, which has strengthened both the clarity and rigor of our study.
Round 2
Reviewer 1 Report
Comments and Suggestions for Authors
Few minor points need to be corrected before acceptance
- Pg5, median baseline eosinophil count was 92.3/uL, the original version was 103/uL, please explain the difference.
- P valu needs to be italicized and use uppercase consistently throughout the manuscript.
- Place table caption of Table 2 above the table
Author Response
1Pg5, median baseline eosinophil count was 92.3/uL, the original version was 103/uL, please explain the difference.
Thank you for pointing this out. In the original submission, we inadvertently reported the median eosinophil count of the entire study cohort in the baseline characteristics table. During the revision process, we corrected this to reflect the actual median baseline eosinophil count used for the analyses within the ICI monotherapy group, which is 92.3/µL.
Please note that all statistical analyses were performed using the correct value (92.3/µL), and therefore the results and conclusions of the study remain unchanged.
2 P value needs to be italicized and use uppercase consistently throughout the manuscript.
Thank you for pointing this out. In accordance with the journal’s formatting guidelines, we have revised the manuscript to ensure that P values are italicized and that the use of uppercase “P” is applied consistently throughout the text, tables, and figure legends.
3 Place table caption of Table 2 above the table
Thank you for pointing this out. The caption for Table 2 has now been moved above the table.
Reviewer 2 Report
Comments and Suggestions for Authors
Thank you for the new analyses.
Author Response
We sincerely appreciate your careful review and your positive feedback on the additional analyses. Please let us know if any further clarification or modifications are needed—we would be happy to address them promptly.